# Blockade of glucagon signaling prevents or reverses diabetes onset only if residual β-cells persist

Nicolas Damond[1,2,3], Fabrizio Thorel[1,2,3], Julie S Moyers[4], Maureen J Charron[5], Patricia M Vuguin[6], Alvin C Powers[7,8], Pedro L Herrera[1,2,3]*

[1]Department of Genetic Medicine and Development of the Faculty of Medicine, University of Geneva, Geneva, Switzerland; [2]Institute of Genetics and Genomics in Geneva, University of Geneva, Geneva, Switzerland; [3]Centre facultaire du diabète, University of Geneva, Geneva, Switzerland; [4]Lilly Research Laboratories, Eli Lilly and Company, Indianapolis, United States; [5]Departments of Biochemistry, Medicine, and Obstetrics & Gynecology and Women's Health, Albert Einstein College of Medicine, Bronx, United States; [6]Pediatric Endocrinology, Women's and Childrens Health, College of Physicians & Surgeons, Columbia University, New York, United States; [7]Division of Diabetes, Endocrinology & Metabolism, Department of Medicine, Department of Molecular Physiology, Vanderbilt University, Nashville, United States; [8]VA Tennessee Valley Healthcare System, Nashville, United States

*For correspondence: Pedro.
Herrera@unige.ch

Competing interest: See
page 14

Reviewing editor: Guy Rutter,
Imperial College London, United
Kingdom

**Abstract** Glucagon secretion dysregulation in diabetes fosters hyperglycemia. Recent studies report that mice lacking glucagon receptor ($Gcgr^{-/-}$) do not develop diabetes following streptozotocin (STZ)-mediated ablation of insulin-producing β-cells. Here, we show that diabetes prevention in STZ-treated $Gcgr^{-/-}$ animals requires remnant insulin action originating from spared residual β-cells: these mice indeed became hyperglycemic after insulin receptor blockade. Accordingly, $Gcgr^{-/-}$ mice developed hyperglycemia after induction of a more complete, diphtheria toxin (DT)-induced β-cell loss, a situation of near-absolute insulin deficiency similar to type 1 diabetes. In addition, glucagon deficiency did not impair the natural capacity of α-cells to reprogram into insulin production after extreme β-cell loss. α-to-β-cell conversion was improved in $Gcgr^{-/-}$ mice as a consequence of α-cell hyperplasia. Collectively, these results indicate that glucagon antagonism could i) be a useful adjuvant therapy in diabetes only when residual insulin action persists, and ii) help devising future β-cell regeneration therapies relying upon α-cell reprogramming.

## Introduction

Glucagon, a 29-amino acid-long hormone synthetized in pancreatic α-cells through cleavage of its precursor, proglucagon, by prohormone convertase 2 (PC2), counterbalances the effects of insulin on blood glucose homeostasis by stimulating hepatic glycogenolysis and gluconeogenesis (*Gromada et al., 2007*). In addition, the two hormones act in a paracrine fashion to reciprocally regulate α- and β-cell function (*Unger and Orci, 2010*).

Hypersecretion of glucagon in diabetes exacerbates hepatic glucose output, thereby fostering hyperglycemia and ketogenesis (*Unger et al., 1970*; *Unger, 1971*; *Sherwin et al., 1976*; *D'Alessio, 2011*). In consequence, antagonists of glucagon signaling are currently being tested in clinical trials for diabetes (*Campbell and Drucker, 2015*). The importance of glucagon signaling in diabetes was

**eLife digest** After meals, digested food causes sugar to accumulate in the blood. This triggers the release of the hormone insulin from beta cells in the pancreas, which allows liver cells, muscle cells and fat cells to use and store the sugar for energy. Other cells in the pancreas, called alpha cells, release a hormone called glucagon that counteracts the effects of insulin by telling the liver to release sugar into the bloodstream. The balance between the activity of insulin and glucagon keeps blood sugar levels steady.

Diabetes results from the body being unable to produce enough insulin or respond to the insulin that is produced, which results in sugar accumulating in the blood. Diabetes also increases the production of glucagon, which further increases blood sugar levels. Recently, some researchers have reported that mice that lack the receptor proteins through which glucagon works do not develop diabetes, even when they are treated with a drug called streptozotocin that wipes out most of their beta cells. This suggests that the high blood sugar levels seen in diabetes result from an excess of glucagon, and not a lack of insulin.

Drugs that block the action of glucagon have been found to reduce the symptoms of mild diabetes in mice and are now being tested in humans. However, it is less clear whether this treatment has any benefits in animals with more severe diabetes.

Streptozotocin destroys most of a mouse's beta cells but a significant fraction of them persist, while a different system relying on diphtheria toxin destroys more than 99% of these cells. Damond et al. have now found that treating mice that lack glucagon receptors with diphtheria toxin causes the mice to develop severe diabetes. Mice that lacked glucagon receptors that had been treated with streptozotocin also developed diabetes after they had been treated with an insulin-blocking drug. Further experiments showed that blocking glucagon receptors in typical mice with diabetes reduces blood sugar, but only if there is some insulin left in their bodies.

Damond et al. also found that the glucagon receptor-lacking mice have more alpha cells, which have the ability to convert into insulin-producing cells after the widespread destruction of beta cells. Together, the experiments suggest that blocking glucagon could be a useful treatment for diabetes, but only in individuals who still have some insulin-producing cells. Such treatment would help reduce the release of sugar from the liver and increase the production of insulin in converted alpha cells in the pancreas. Damond et al. are now investigating how alpha cells convert into beta cells, with the aim of learning how to make beta cells regenerate more efficiently.

recently highlighted in studies performed with glucagon receptor knockout (*Gcgr$^{-/-}$*) mice and in animals lacking α-cells due to pancreatic aristaless-related homeobox (*Arx*) deficiency. Surprisingly, these animals did not exhibit the usual signs of diabetes, such as hyperglycemia or glucose intolerance, after streptozotocin (STZ)-mediated β-cell destruction (*Conarello et al., 2006*; *Lee et al., 2011*; *2012*; *Hancock et al., 2010*). These findings lead to hypothesize that glucagon is responsible for the features of diabetes (*Unger and Cherrington, 2012*). Although suppression of glucagon action is likely to attenuate the consequences of insulin deficiency, its primary role in the hyperglycemia is uncertain. Indeed, because STZ causes an incomplete β-cell ablation due to variations in administration protocols and in genetic background-dependent sensitivity (*Deeds et al., 2011*; *Cardinal et al., 1998*; *Gurley, 2006*), it is possible that the "diabetes resistance" phenotype of *Gcgr$^{-/-}$* mice relies on the action of insulin from residual β-cells. Thus, to determine whether lack of glucagon signaling would also prevent hyperglycemia and diabetes in the context of a more severe insulin deficiency, we used a transgenic model of diphtheria toxin (DT)-mediated β-cell ablation, termed *RIP-DTR*, which leads to an almost complete β-cell elimination (*Thorel et al., 2010*; *Chera et al., 2014*). Also, because adult *RIP-DTR* mice spontaneously reconstitute new insulin-producing cells by α-cell transdifferentiation in this condition of severe insulin insufficiency, we explored whether the compensatory α-cell hyperplasia due to glucagon signaling blockade (*Furuta et al., 1997*; *Gelling et al., 2003*; *Longuet et al., 2013*) influences the reprogramming of α-cells toward insulin production.

Here we show that near-total β-cell loss triggers severe hyperglycemia and all the metabolic features of type 1 diabetes (cachexia, glucose intolerance, and death) in mice with constitutive or

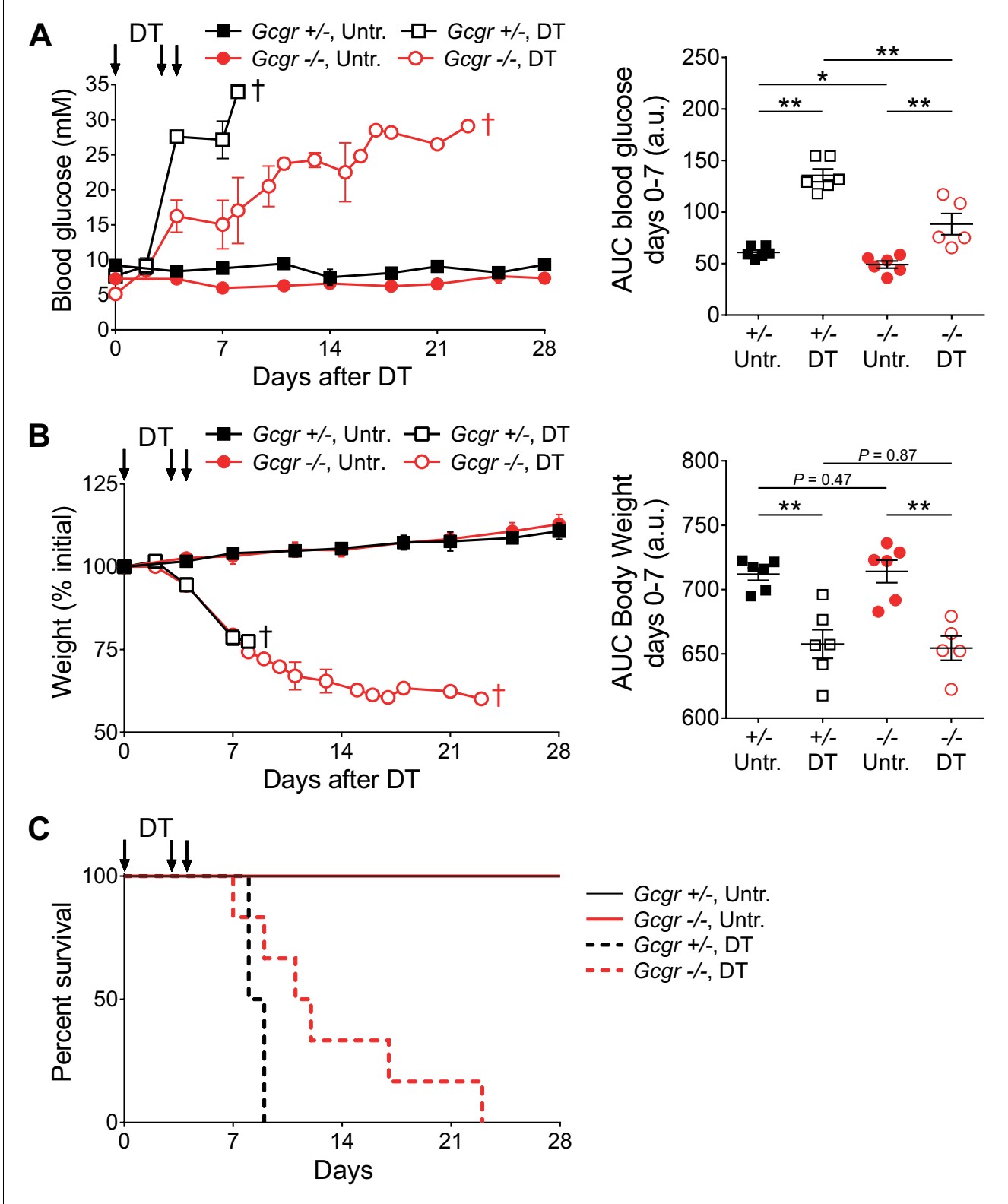

**Figure 1.** *Gcgr*[-/-] mice become diabetic after massive β-cell ablation. (**A**) Random-fed glycemia (*left*) and area under the glycemia curve (AUC) between days 0 and 7 after DT (*right*) in untreated (Untr.) and DT-treated *RIP-DTR;Gcgr*[+/-] and *RIP-DTR;Gcgr*[-/-] females. (**B**) Body weight (*left*) and AUC body weight (days 0–7 after DT; *right*). †, all mice of the group were dead at this time point (see *Figure 1C*). *p<0.05; **p<0.01; Mann-Whitney *U* test. C: *Figure 1 continued on next page*

*Figure 1 continued*

Survival curve of *RIP-DTR;Gcgr⁺/⁻* and *RIP-DTR;Gcgr⁻/⁻* mice after DT treatment (N=5–6). Survival analysis of DT-treated animals (*Gcgr⁺/⁻* versus *Gcgr⁻/⁻*): p=0.044; Log-rank test.

The following figure supplement is available for figure 1:

**Figure supplement 1.** Insulin administration stabilizes body weight and allows survival of DT-treated *Gcgr⁻/⁻* mice.

induced glucagon signaling deficiency. We report that the absence of hyperglycemia observed in glucagon-deficient mice after STZ treatment can be explained through the persistence of a residual β-cell mass, which ensures a low level of insulin action.

## Results

### Near-total β-cell ablation leads to full-blown diabetes in mice lacking glucagon signaling

Recent reports indicate that *Gcgr⁻/⁻* mice do not develop hyperglycemia after STZ-mediated β-cell loss. Here we aimed at determining the effect of the absence of glucagon action in the context of a more extreme insulin deficiency. For this purpose, we crossed *Gcgr⁻/⁻* mutant animals (*Gelling et al.,*

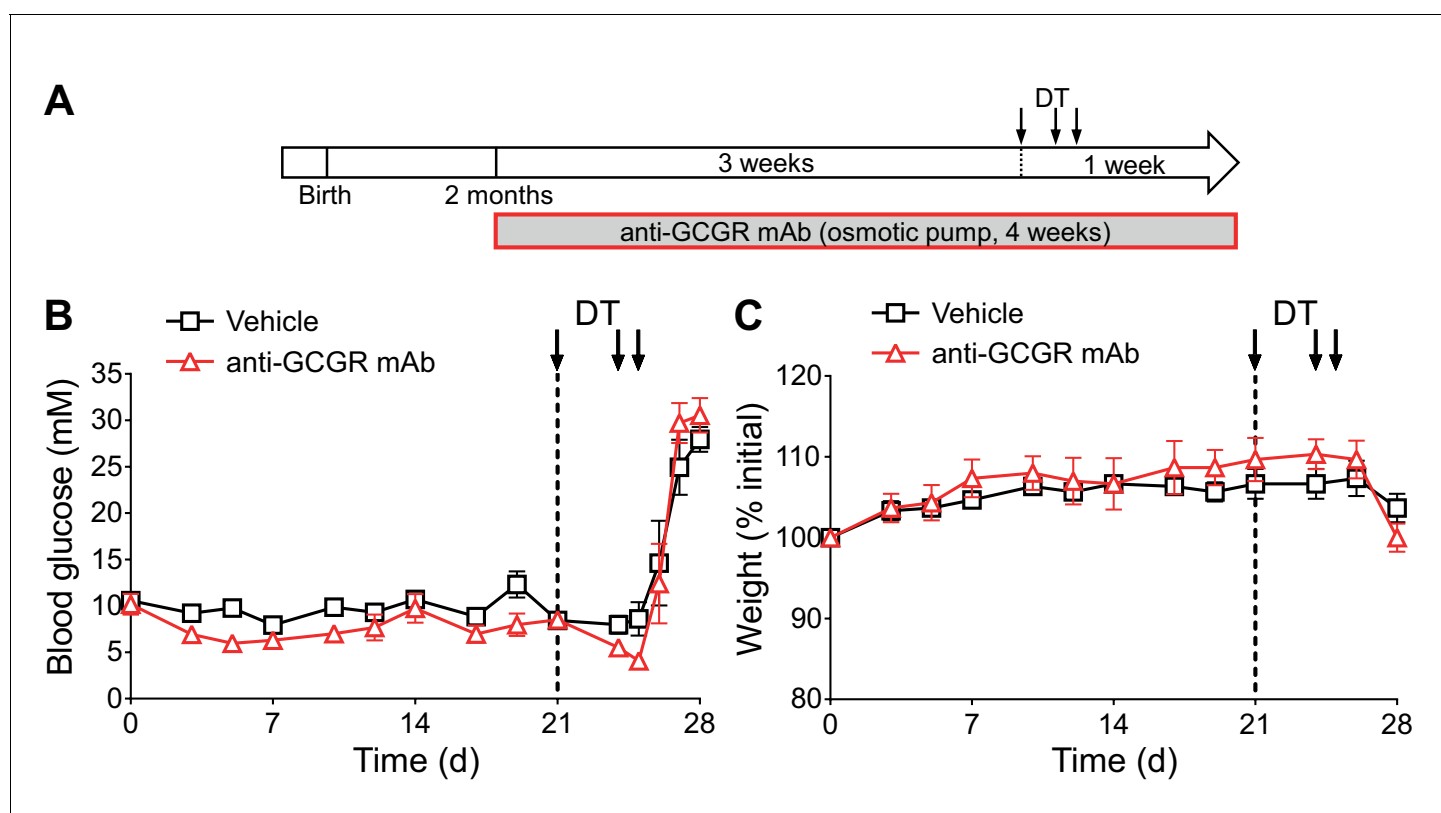

**Figure 2.** Anti-GCGR mAb-treated mice become diabetic after massive β-cell ablation. (**A**) Experimental design. (**B-C**) Random-fed glycemia (**B**) and body weight (**C**) after DT in C57BL/6 males pre-treated with vehicle or mAb (N=3).

The following figure supplements are available for figure 2:

**Figure supplement 1.** Anti-GCGR mAb administration recapitulates the metabolic and cellular phenotypes of *Gcgr⁻/⁻* mice.

**Figure supplement 2.** Insulin administration is required to stabilize body weight and allow survival of anti-GCGR-treated mice after DT.

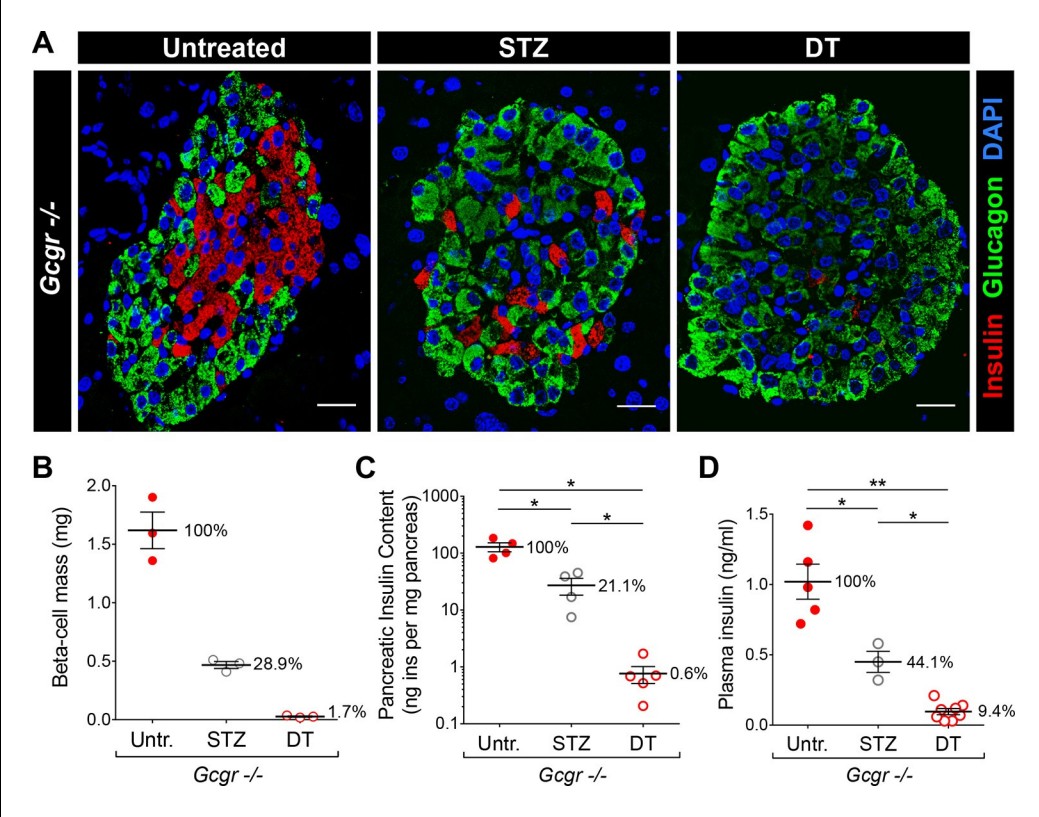

**Figure 3.** DT administration leads to a more complete β-cell ablation than STZ. (A) Islet sections stained for insulin (red) and glucagon (green) from untreated, STZ-, or DT-treated *RIP-DTR;Gcgr⁻/⁻* females, 6 days after the last STZ or DT injection. Scale bars: 20 μm. (B-D) β-cell mass (B), pancreatic insulin content (C) and fed plasma insulin levels (D) in untreated (Untr.), STZ-, or DT-treated *RIP-DTR;Gcgr⁻/⁻* males and females, 6 days after the last injection. STZ administration: two injections (200 and 150 mg/kg). *p<0.05; **p<0.01; Mann-Whitney *U* test.

The following figure supplements are available for figure 3:

**Figure supplement 1.** *RIP-DTR;Gcgr⁻/⁻* mice remain hyperglucagonemic and α-cell mass is not affected after STZ- or DT-treatment.

**Figure supplement 2.** Higher efficiency of β-cell ablation after DT- than after STZ-treatment in mice with normal glucagon signaling.

---

*2003*) with *RIP-DTR* mice, in which diphtheria toxin (DT) injection triggers the near-total (>99% ) β-cell loss (*Thorel et al., 2010*).

*RIP-DTR;Gcgr⁻/⁻* mice, like *Gcgr⁻/⁻* mice, displayed lower basal glucose levels than controls (*RIP-DTR;Gcgr⁺/⁺* and *RIP-DTR;Gcgr⁺/⁻*; not shown) (*Gelling et al., 2003*). Upon DT-induced β-cell ablation, both control and knockout animals developed severe hyperglycemia, with a slower kinetics in *RIP-DTR;Gcgr⁻/⁻* mice (*Figure 1A*). Animals of both groups lost weight at similar rates (*Figure 1B*), and died in absence of exogenous insulin treatment (*Figure 1C*). By contrast, administration of long-acting insulin, although insufficient to normalize blood glucose levels, permitted survival and body weight maintenance (*Figure 1—figure supplement 1*). As soon as insulin treatment was discontinued, blood glucose levels and body weight quickly deteriorated in all groups. Altogether, these findings indicate that *Gcgr⁻/⁻* mice are not protected against hyperglycemia after near-total β-cell loss, but develop classical signs of type 1 diabetes and require insulin therapy.

Constitutive *Gcgr* deletion leads to increased embryonic lethality, and defects in pancreatic development and islet-cell maturation (*Vuguin et al., 2006*; *Vuguin and Charron, 2011*; *Ouhilal et al., 2012*). Since these abnormalities may encompass long-lasting compensatory

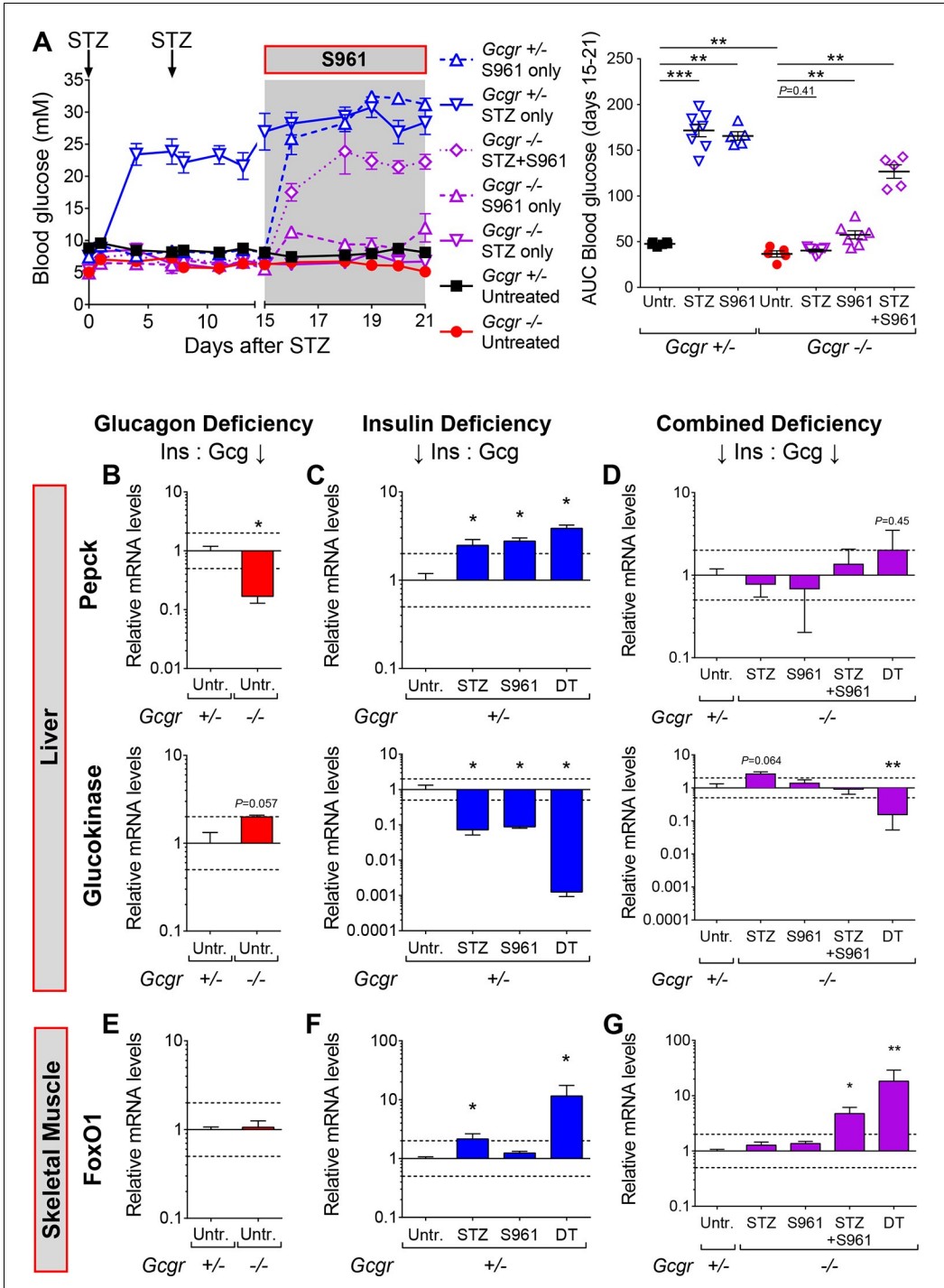

**Figure 4.** Inhibition of insulin action triggers hyperglycemia in STZ-treated $Gcgr^{-/-}$ mice. (**A**) Random-fed glycemia after STZ and/or S961 administration in $Gcgr^{+/-}$ and $Gcgr^{-/-}$ females (left), and area under the glycemia curve (AUC) during S961 treatment (right). (**B-D**) Hepatic Pepck (top) and Glucokinase (bottom) mRNA levels relative to those of untreated $Gcgr^{+/-}$ (control) mice (N=4–6). (**B**) Glucagon deficiency: $Gcgr^{-/-}$ background. (**C**) Insulin deficiency: β-cell ablation or insulin signaling inhibition. (**D**) Combined deficiency: β-cell ablation and/or insulin signaling inhibition in a $Gcgr^{-/-}$ background. (**E-G**) FoxO1 mRNA levels in skeletal muscle, relative to those of untreated $Gcgr^{+/-}$ mice (N=4–6). STZ administration: 200 mg/kg at day 0 and 150 mg/kg at day 7. S961 treatment: osmotic pump (days 15 to 21). *p<0.05; **p<0.01; Mann-Whitney $U$ test. Only groups that exhibited a > twofold regulation as compared to controls (dashed lines) were tested.

The following figure supplements are available for figure 4:

*Figure 4 continued on next page*

*Figure 4 continued*

**Figure supplement 1.** Higher hepatic PEPCK protein expression after DT in both *Gcgr^+/-* and *Gcgr^-/-* mice.

**Figure supplement 2.** Liver glycogen concentration is reduced after DT-treatment in both *RIP-DTR-Gcgr^+/-* and *RIP-DTR-Gcgr^-/-* mice.

**Figure supplement 3.** Expression of genes negatively regulated by insulin signaling in skeletal muscle.

metabolic adaptations, we conditionally inhibited glucagon action in adult mice that had developed normally using a glucagon receptor antagonizing monoclonal antibody (anti-GCGR mAb). We first assessed its activity in C57BL/6 wild type mice (*Figure 2—figure supplement 1A*). In agreement with a previously described antibody (*Gu et al., 2009*; *Yan et al., 2009*), anti-GCGR treatment led to a reduction in basal glycemia (*Figure 2—figure supplement 1B*), and triggered α-cell hyperplasia and hypertrophy, as observed in *Gcgr^-/-* animals (*Figure 2—figure supplement 1C–D*) (*Gelling et al., 2003*). In addition, antibody-treated *Gcgr^+/+* mice showed altered responses, like *Gcgr^-/-* animals, to intraperitoneal glucose and insulin tolerance tests (*Figure 2—figure supplement 1E–F*). Anti-GCGR administration in *Gcgr^+/+* mice therefore phenocopies the main metabolic and cellular alterations of *Gcgr^-/-* mice and thus represents a valuable tool for inducing glucagon signaling antagonism in vivo.

To assess whether induced glucagon receptor blockade prevents diabetes upon near-total β-cell ablation, we pre-treated adult *RIP-DTR* mice with the anti-GCGR mAb for 3 weeks, and then injected them with DT (*Figure 2A*). In agreement with the above results using *RIP-DTR;Gcgr^-/-* animals, all mice became severely hyperglycemic and lost weight after DT, regardless of antibody treatment (*Figure 2B–C*). Moreover, only insulin administration allowed for survival following β-cell ablation, not glucagon receptor inhibition (*Figure 2—figure supplement 2*). Collectively, these observations indicate that the lack of glucagon signaling is not sufficient per se to prevent severe hyperglycemia and diabetes following extreme β-cell loss, and contrast with previous studies in which *Gcgr^-/-*, or anti-GCGR-treated mice did not develop the metabolic manifestations of the disease when β-cell ablation was mediated by STZ (*Conarello et al., 2006*; *Lee et al., 2011*; *2012*; *Wang et al., 2015*).

## DT leads to a more complete β-cell ablation than STZ

The different impact of STZ and DT treatments on glycemia in *Gcgr^-/-* mice may result from a difference in completeness of β-cell destruction. To test this hypothesis, we compared the relative ablation efficiencies of these two methods. To maximize β-cell destruction, we treated *Gcgr^+/-* and *Gcgr^-/-* mice with two high doses of STZ (200 and 150 mg/kg, one week apart). Following the first injection, control mice became severely hyperglycemic. By contrast, *Gcgr^-/-* animals remained normoglycemic even after the second STZ injection, as previously reported (not shown) (*Lee et al., 2011*; *2012*). *RIP-DTR;Gcgr^-/-* animals remained markedly hyperglucagonemic after STZ- or DT-mediated β-cell loss and α-cell mass was not affected (*Figure 3—figure supplement 1A–B*). Histologically, we observed that nearly 90% of islet sections were totally devoid of β-cells after DT, versus only 45% after STZ (*Figure 3A*). Accordingly, the β-cell mass and pancreatic insulin content were reduced by 98–99% after DT, but only by 70–80% after STZ (*Figure 3B–C*). In addition, plasma insulin levels were just above detection threshold after DT, but readily detectable after STZ (*Figure 3D*). We made similar observations in mice with normal glucagon signaling (*Figure 3—figure supplement 2*). Together, these results indicate that β-cell destruction is more complete after DT- than after STZ-treatment in *Gcgr^-/-* mice.

## Residual insulin action protects STZ-treated *Gcgr^-/-* mice from hyperglycemia

Because β-cell ablation was incomplete after STZ, we aimed at determining whether the action of residual circulating insulin might, in combination with glucagon signaling deficiency, protect *Gcgr^-/-* mice from diabetes.

To test this hypothesis, we inhibited insulin action using the insulin receptor antagonist drug S961 (*Schäffer et al., 2008*). In vivo, S961 administration induces hyperglycemia in wild type animals and closely recapitulates the phenotype of mice with liver-specific insulin receptor deletion (*Yi et al.,*

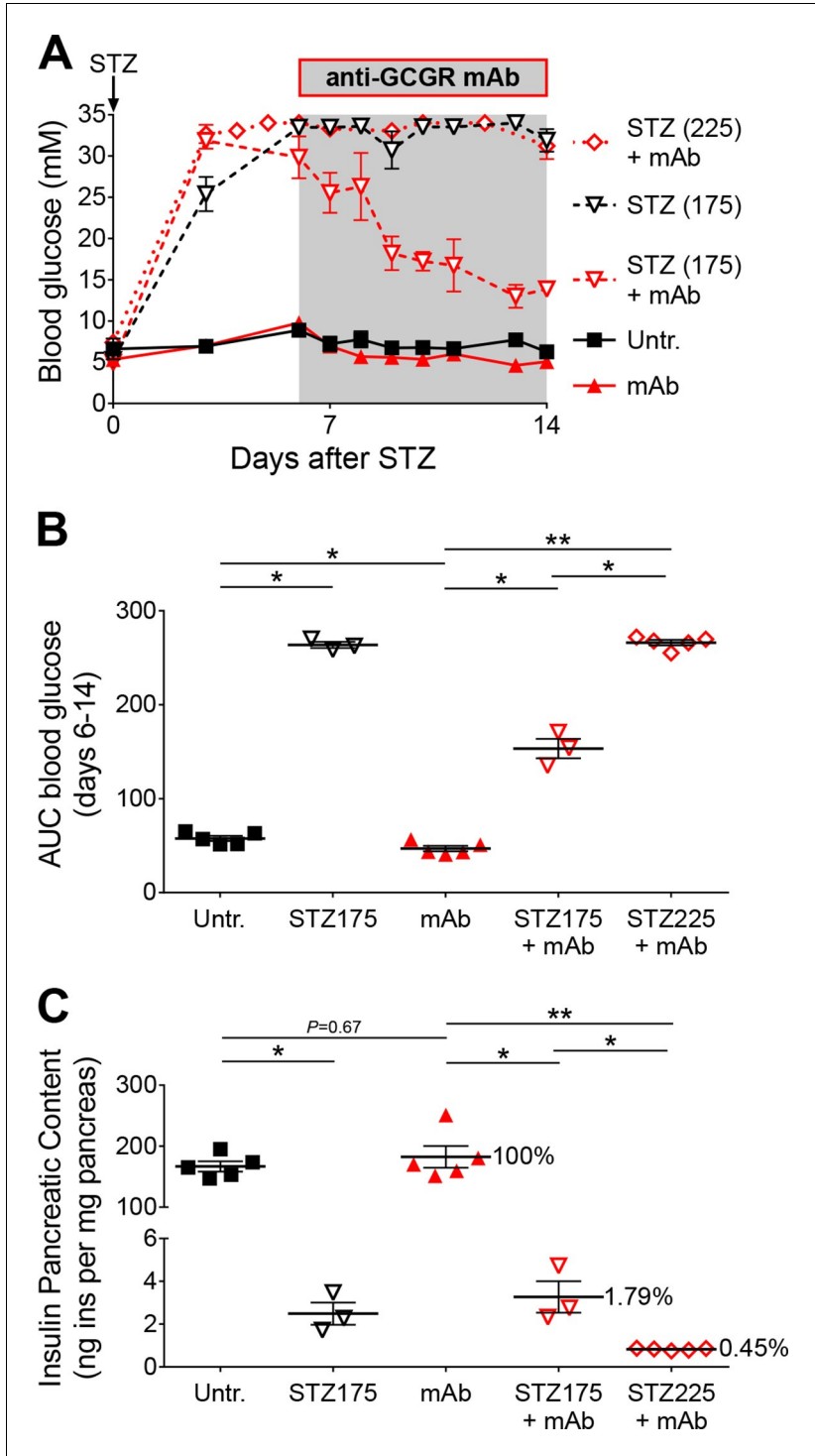

**Figure 5.** Anti-GCGR mAb treatment does not normalize hyperglycemia after efficient STZ-mediated β-cell ablation. (**A**) Random-fed glycemia in C57BL/6 males treated with STZ (single injection at day 0: 175 or 225 mg/kg) and/or anti-GCGR mAb (osmotic pump, days 6 to 14; N=3–6). (**B**) Area under the glycemia curves during mAb treatment. (**C**) Pancreatic insulin content. *p<0.05; **p<0.01; Mann Whitney *U* test.

The following figure supplement is available for figure 5:

**Figure supplement 1.** Hepatic Pepck and Glucokinase expression after STZ and/or anti-GCGR mAb treatment.

2013; *Michael et al., 2000*). In agreement with its previously reported action, S961 administration in *Gcgr^+/-* mice triggered a strong increase in glycemia (*Figure 4A*; blue dashed vs black continuous line). Interestingly, *Gcgr^-/-* animals exhibited a smaller but significant increase in glycemia, indicating that glucagon deficiency has a beneficial effect in this situation of relative insulin deficit (purple dashed vs red continuous line). Although STZ-treated *Gcgr^-/-* mice remained normoglycemic, as previously reported (*Conarello et al., 2006*; *Lee et al., 2011*; *2012*), they developed severe hyperglycemia after insulin receptor inhibition (continuous vs dotted purple line). This suggests that residual insulin action, likely originating from STZ-escaping β-cells, is still present after STZ administration in *Gcgr^-/-* animals, and is necessary to prevent hyperglycemia and diabetes.

To better characterize the effect of insulin insufficiency in a glucagon-deficient context, we evaluated hepatic transcript levels of Phosphoenolpyruvate carboxykinase (Pepck) and Glucokinase (Gck), two hormone-sensitive enzymes whose transcription is regulated by the relative levels of glucagon and insulin signaling (*Rucktäschel et al., 2000*; *Chakravarty et al., 2005*; *Iynedjian et al., 1995*). Liver is a relevant organ to assess the impact of insulin and glucagon deficiency because re-expression of the glucagon receptor in the liver of STZ-treated *Gcgr^-/-* mice, and conditional inactivation of the insulin receptor in hepatocytes are both sufficient to trigger hyperglycemia (*Lee et al., 2012*;

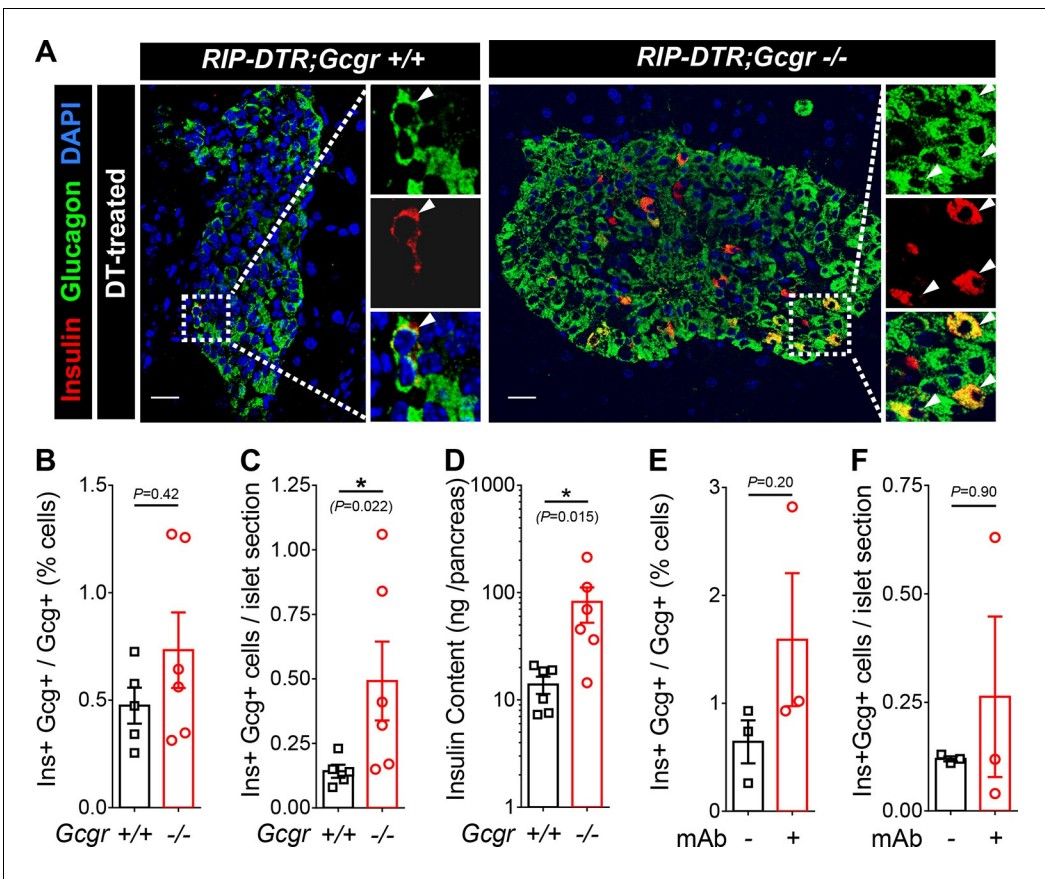

**Figure 6.** Absence of glucagon signaling does not block the appearance of new glucagon-insulin bihormonal cells after β-cell ablation. (A) Islet sections exhibiting glucagon-insulin co-expressing cells (arrowheads) from *RIP-DTR; Gcgr^+/+* and *RIP-DTR;Gcgr^-/-* females (1 m after DT). Scale bars: 20 μm. (B-D) Percentage of glucagon[+] cells that co-express insulin (B), bihormonal cells per islet section (C), and pancreatic insulin content (D) in *RIP-DTR;Gcgr^+/+* and *RIP-DTR;Gcgr^-/-* females (1 m after DT, N=5–6). (E-F) Percentage of glucagon[+] cells that co-express insulin (E), and bihormonal cells per islet section (F) in vehicle- or anti-GCGR mAb- treated *RIP-DTR* males (2 weeks after DT, N=3). *p<0.05; Mann-Whitney *U* test.

The following figure supplement is available for figure 6:

**Figure supplement 1.** Newly formed bihormonal cells in *Gcgr^-/-* mice are reprogrammed α-cells.

*Michael et al., 2000*). In conditions of glucagon deficiency (increased insulin/glucagon ratio; *Gcgr*[-/-] mice), we observed a decreased expression of the gluconeogenic enzyme Pepck and an increased expression of the glycolytic enzyme Gck as compared to *Gcgr*[+/-] controls (*Figure 4B*), which is consistent with a previous study (*Yang et al., 2011*). By contrast, upon induced insulin deficiency (decreased insulin/glucagon ratio), as in STZ-, S961-, or DT-treated *Gcgr*[+/-] animals, Pepck and Gck exhibited the opposite regulation (*Figure 4C*). We observed the strongest effect after DT, which caused a 1000-fold decrease in Gck expression, suggesting that it led to a more complete suppression of insulin action than STZ or S961. When inducing insulin insufficiency in a *Gcgr*[-/-] background, a situation of combined insulin and glucagon deficiency, we observed Pepck and Gck mRNA levels similar to those measured in untreated *Gcgr*[+/-] control mice, except after DT, which induced a strong downregulation of Gck expression in *Gcgr*[-/-] livers (*Figure 4D*). We also confirmed the increase in hepatic PEPCK expression after DT at the protein level (*Figure 4—figure supplement 1*). Similarly, DT-, but not STZ-treatment depleted liver glycogen stores in *RIP-DTR;Gcgr*[-/-] animals (*Figure 4—figure supplement 2*). These results suggest that lack of glucagon action can compensate for the effect of partial insulin insufficiency on the expression of rate-limiting enzymes and hepatic glycogen metabolism, but not after near-total β-cell loss, a situation where the effect of insulin deficiency outweighs that of glucagon deficiency.

We then assessed insulin signaling activity in skeletal muscle by measuring the expression of the transcription factor Forkhead box protein O1 (FoxO1) and of several of its target genes, such as Insulin receptor substrate 2 (Irs2), which are induced upon insulin insufficiency (*Long et al., 2011*). FoxO1 mRNA levels were similar in untreated *Gcgr*[+/-] and *Gcgr*[-/-] mice (*Figure 4E*). In *Gcgr*[-/-] animals, STZ or S961 administration did not significantly affect FoxO1 expression. By contrast, FoxO1 and its targets were strongly upregulated upon combined STZ and S961-, or DT-treatment, reflecting a more severe insulin insufficiency (*Figure 4G* and *Figure 4—figure supplement 3*).

Together, these results indicate that lack of glucagon signaling efficiently compensates for the consequences of insulin insufficiency only if residual insulin action persists after β-cell loss.

## Glucagon signaling blockade attenuates hyperglycemia after STZ-mediated β-cell loss only when residual insulin production persists

As *Gcgr*[-/-] mice exhibit resistance to STZ-induced hyperglycemia, we assessed the impact of glucagon signaling blockade on C57BL/6 mice made hyperglycemic with a single injection of either 175 or 225 mg/kg STZ. Once the animals were hyperglycemic, we implanted them with an osmotic pump containing the anti-GCGR mAb. In mice injected with 175 mg/kg STZ, antibody treatment strongly reduced, but did not completely normalize, blood glucose levels (*Figure 5A and B*). By contrast, animals that had received 225 mg/kg STZ remained severely hyperglycemic (>30 mM) after anti-GCGR mAb administration. As expected, residual pancreatic insulin content negatively correlated with the dose of STZ (*Figure 5C*). We thus observed beneficial effects of glucagon signaling inhibition only in diabetic mice that had retained a relatively higher pancreatic insulin after STZ-mediated β-cell loss. Strikingly, the impact of glucagon signaling inhibition on the glycemia of diabetic mice was dependent on very small measurable differences in residual pancreatic insulin, as seen after 175 and 225 mg/kg STZ (respectively 1.79% and 0.45% of the pancreatic insulin content of non-ablated controls). As seen in *Gcgr*[-/-] animals, anti-GCGR mAb administration resulted in a lower expression of hepatic Pepck (*Figure 5—figure supplement 1*). In addition, the highest STZ dose triggered a stronger glucokinase downregulation than the 175 mg/kg dose in mAb-treated mice.

Collectively, our findings support the notion that, regardless the method of β-cell ablation (STZ or DT), the beneficial effects of inhibiting glucagon action, either genetically or pharmacologically, rely upon residual insulin action.

## Induction of insulin production in α-cells after β-cell ablation also occurs in absence of glucagon signaling

We have previously shown that massive β-cell ablation triggers insulin expression in a small fraction of the α-cell population, with the appearance of glucagon/insulin bihormonal cells (*Thorel et al., 2010*). We report above that in such a situation of near-total β-cell loss, lack of glucagon action fails to normalize glycemia. We then assessed whether the α-cell expansion triggered by glucagon signaling inhibition could have a beneficial effect on α-cell reprogramming. One month after DT-mediated

β-cell ablation, we observed bihormonal cells in *RIP-DTR;Gcgr^+/+* and *RIP-DTR;Gcgr^-/-* mice (*Figure 6A*). Because *RIP-DTR;Gcgr^-/-* animals have α-cell hyperplasia (*Gelling et al., 2003*; *Longuet et al., 2013*) and the number of bihormonal cells was proportional to the number of α-cells in both groups (*Figure 6B*), we observed a significant increase in the absolute number of bihormonal cells in *RIP-DTR;Gcgr^-/-* mice (*Figure 6C*). Consistent with these observations, they had a higher pancreatic insulin content (*Figure 6D*). These results indicate that there is an increased number of α-cells engaged into reprogramming in mice lacking glucagon signaling. We also observed the appearance of bihormonal cells in DT-treated adult *RIP-DTR* mice undergoing anti-GCGR mAb treatment (*Figure 6E–F*). We confirmed the α-cell origin of these newly formed bihormonal cells using a previously described tetracycline-activated system, which allows the specific and efficient doxycycline (DOX)-dependent irreversible tracing of α-cells with YFP (*Figure 6—figure supplement 1A–B*) (*Thorel et al., 2010*). One month after DT injection in *Gcgr^-/-* mice, we observed that a significant fraction of insulin-producing cells were also YFP-positive and therefore derived from cells that had previously expressed glucagon (*Figure 6—figure supplement 1C*). We confirmed these observations in animals in which conditional GCGR inhibition was applied after DT-mediated β-cell ablation (*Figure 6—figure supplement 1D*).

Together, these findings indicate that although glucagon signaling blockade does not prevent hyperglycemia in diabetic mice that exhibit extreme insulin deficiency, it results in enhanced formation of new insulin-producing cells by increasing the absolute number of converting α-cells.

## Discussion

Glucagon receptor inhibition decreases hyperglycemia in various animal models of diabetes (*Gu et al., 2009*; *Johnson et al., 1982*; *Brand et al., 1994*; *Sloop et al., 2004*; *Mu et al., 2011*; *Sorensen et al., 2006*), as well as in patients with type 2 diabetes (*Kelly et al., 2015*). The extent of these benefits remains however disputed in situations where the β-cell population is nearly completely depleted, as in long-standing type 1 diabetes (*Wang et al., 2012*; *Meier et al., 2005*). Previous studies have shown that STZ-mediated β-cell ablation does not induce diabetes in the *Gcgr^-/-* mouse model (*Conarello et al., 2006*; *Lee et al., 2011*; *2012*), giving rise to the hypothesis that mice cannot develop hyperglycemia in absence of glucagon action (*Unger and Cherrington, 2012*). Here, we show that *Gcgr^-/-* and anti-GCGR mAb-treated animals develop severe hyperglycemia after massive DT-mediated β-cell ablation (*Figures 1* and *2*). Our results suggest that the disparity in blood glucose levels observed between STZ- and DT-treated *Gcgr^-/-* animals originate from a difference in β-cell destruction efficiency (*Figure 3*).

Recent studies reached conflicting conclusions regarding the beneficial effect of glucagon signaling blockade in severely diabetic mice: Wang et al reported that anti-GCGR mAb treatment was sufficient to normalize glycemia of STZ-treated BALB/c animals (*Wang et al., 2015*), whereas Steenberg et al did not observe improvements in glucose tolerance after GCGR antagonism or glucagon immunoneutralisation in C57BL/6 mice (*Steenberg et al., 2016*). These discrepancies may be explained by differences in completeness of β-cell ablation linked to the protocol of injection (single high dose versus multiple low doses) and/or to strain-dependent sensitivity; it was indeed reported that BALB/c mice are less sensitive to STZ than C57BL/6 animals (*Cardinal et al., 1998*; *Gurley, 2006*). Here, we injected C57BL/6 mice with two different high doses of STZ that triggered a severe hyperglycemia; after anti-GCGR mAb treatment, however, we observed a decrease in glycemia only in animals treated with the lowest STZ dose. These results indicate that a small difference in pancreatic insulin, such as that observed after 175 and 225 mg/kg STZ, can cause a major difference in glycemia in animals lacking glucagon signaling, thereby highlighting the importance of residual insulin action and providing a potential explanation for discrepancies between previous studies (*Figure 5*). Remarkably, STZ-treated *Gcgr^-/-* mice became hyperglycemic upon S961-mediated insulin receptor antagonism, illustrating the requirement of residual insulin action for maintenance of normoglycemia in these animals (*Figure 4*). Collectively, these findings demonstrate that a total absence of glucagon action is not sufficient to prevent hyperglycemia in case of severe insulin deficiency.

Although *Gcgr^-/-* mice developed diabetes upon massive β-cell ablation, lack of glucagon action reduced or normalized glycemia in conditions of less severe insulin deficiency. In particular we observed that i) anti-GCGR mAb administration reduced hyperglycemia in C57BL/6 mice treated

with the lowest STZ dose (*Figure 5*) and ii) S961 treatment caused a less severe increase in glycemia in *Gcgr^-/-* than in *Gcgr^+/-* animals (*Figure 4A*). Our data on hepatic expression of Pepck and Gck, two rate-limiting enzymes of gluconeogenesis and glycolysis, respectively, suggest that lack of glucagon signaling counterbalances the effects of insulin insufficiency after STZ or S961. This would prevent, or limit, the rise in net hepatic glucose output by decreasing gluconeogenesis and glycogenolysis, and by increasing glycolysis and glycogenesis. Absence of glucagon action is however not sufficient to compensate severe insulin deficiency after DT, as reflected by Gck downregulation and reduced hepatic glycogen content, thereby contributing to the elevation of blood glucose. Interestingly, the mRNA levels of FoxO1 target genes in skeletal muscle were strongly upregulated, reflecting insulin signaling insufficiency, after DT and STZ+S961, the two conditions that led to hyperglycemia (*Figure 4* and *Figure 4—figure supplement 3*). In addition, gonadal adipose tissue was markedly depleted in experimental conditions leading to hyperglycemia (in *Gcgr^-/-* mice after STZ+S691 and DT; in *Gcgr^+/-* mice after STZ and DT; not shown). Together, these findings provide new insights into the mechanisms by which lack of glucagon signaling protects against elevated blood glucose levels in situations of insulin insufficiency. Recent studies have shown that protection against STZ-mediated hyperglycemia also rely on the high levels of circulating glucagon-like peptide-1 (GLP-1) in *Gcgr^-/-* animals (*Gu et al., 2010*; *Ali et al., 2011*; *Jun, 2014*; *Omar et al., 2014*). Yet, these high levels of GLP-1 combined with a lack of glucagon action were insufficient to maintain normoglycemia after near-total β-cell loss.

Finally, we report here that lack of glucagon signaling does not compromise the ability of α-cells to convert to insulin production after DT-mediated near-total β-cell loss. Indeed, YFP-traced α-cells become glucagon/insulin bihormonal cells after DT in *RIP-DTR;Gcgr^-/-* mice and in animals treated with the anti-GCGR antibody (*Figure 6* and *Figure 6—figure supplement 1*). The proportion of α-cells co-expressing insulin after DT is comparable between mice with either intact, reduced or absent glucagon signaling, indicating that glucagon does not play an essential role in the α-to-β transdifferentiation process. Interestingly, because glucagon signaling inhibition leads to a compensatory α-cell hyperplasia (*Furuta et al., 1997*; *Gelling et al., 2003*; *Longuet et al., 2013*), the absolute number of newly formed insulin-producing cells through α-cell conversion was augmented in *RIP-DTR; Gcgr^-/-* mice. As previously described in adult mice (*Chera et al., 2014*), we also observed the δ-to-β conversion in β-cell-ablated *RIP-DTR;Gcgr^-/-* mice (not shown).

In conclusion, although inhibition of glucagon action alone is insufficient to prevent diabetes in conditions of near-total insulin deficiency, it is beneficial when residual insulin action persists, as in STZ-treated *Gcgr^-/-* animals. Combination of glucagon inhibition with insulin therapy may however increase the risk of hypoglycemia. We encountered this problem when using subcutaneous insulin pellets in DT-treated *RIP-DTR;Gcgr^-/-* mice: they became hypoglycemic and died likely as a consequence of the constitutive insulin release from the pellets, which could not be compensated by glucagon action. Our findings suggest that diabetes therapy through glucagon suppression would be unsafe if exogenous insulin has to be supplemented, but may be beneficial in patients with sufficient residual insulin action. In case of near-total insulin deficiency, transient glucagon receptor blockade could also serve as a means to increase the α-cell mass before triggering insulin production in these cells, a strategy that might be envisioned as a novel therapy to treat diabetes.

## Materials and methods

### Mice

*Gcgr^-/-* (*Gelling et al., 2003*), *RIP-DTR* (Rat insulin promoter - diphtheria toxin receptor) (*Thorel et al., 2010*), *Gcg-rtTA* (Glucagon promoter - reverse tetracycline transactivator) (*Thorel et al., 2010*), *TetO-Cre* (Tetracycline operator - Cre recombinase) (*Perl et al., 2002*), and *R26-YFP* (Rosa26 promoter - yellow fluorescent protein) (*Srinivas et al., 2001*) mice were described previously and bred on a C57BL/6-enriched mixed genetic background. As pups born from *Gcgr^-/-* mothers die perinatally (*Vuguin et al., 2006*), *Gcgr^+/-* females were used for breeding. C57BL/6 mice were purchased from Janvier Labs (France). All mice used in this study were adult (10–20 week old) males or females. They were housed and treated in accordance with the guidelines and regulations of the Direction Générale de la Santé, state of Geneva. Blood glucose was measured from tail

blood using a handheld glucometer (detection range: 0.6 to 33.3 mM, values exceeding 33.3 mM were artificially set to 34 mM).

## Diphtheria toxin (DT), Streptozotocin (STZ), and Doxycycline (DOX) treatments

For β-cell ablation in *RIP-DTR* mice, DT (D0564, Sigma, St. Louis, MO) was injected i.p. in 3 injections of 125 ng each, at days 0, 3, and 4. STZ (S0130, Sigma) was used as an alternative method of β-cell ablation. It was freshly diluted in citrate buffer and administered in 5-h fasted mice. Two different protocols were used depending on the genetic background: i) *Gcgr$^{+/-}$* and *Gcgr$^{-/-}$* mice: two i.p. injections of 200 and 150 mg/kg, one week apart; ii) C57BL/6 mice: single i.p. injection (175 or 225 mg/kg). For inducible α-cell labeling in *Gcg-rtTA;TetO-Cre;R26-YFP* mice, DOX (D9891, Sigma) was added to drinking water (1 mg/ml) for 2 weeks followed by at least 2 weeks of clearance before DT injection.

## Anti-GCGR mAb

Anti-GCGR monoclonal antibody A-9 was generated at Eli Lilly and Company (Yan H, Hu S-FS, Boone TC, Lindberg RA, inventors; Amgen Inc., assignee. Compositions and methods relating to glucagon receptor antibodies. United States patent US 8158759 B2, 2012 Apr 17). It was delivered either via i.p. injections, thrice weekly (9 mg/kg per injection), or using a s.c. implanted osmotic pump (model 2002, Alzet, Cupertino, CA) containing 11 mg/ml of anti-GCGR mAb in PBS (estimated delivery rate: 5.5 µg/h for 2 weeks).

## S961

The insulin receptor inhibitor S961 was a kind gift of Lauge Schäffer (Novo Nordisk, Denmark) (*Schäffer et al., 2008*). Mice were implanted s.c. with an osmotic pump (model 1007D, Alzet) loaded with 40 nmol S961 (estimated delivery rate: 0.25 nmol/h for 1 week).

## Insulin

Long-acting insulin detemir (Levemir, Novo Nordisk) was freshly diluted in NaCl 0.9% and injected s.c. twice per day (1.7 U/kg in the morning, 3.3 U/kg in the evening). Insulin pellets (LinShin Canada Inc., Canada) were implanted s.c.

## Intraperitoneal glucose tolerance test (ipGTT) and insulin tolerance test (ITT)

For the ipGTT, mice were fasted overnight (15 hr) and then injected i.p. with 2 mg/kg *D*-glucose. For the ITT, mice were fasted for 5 hr and injected i.p. with 0.7 U/kg insulin (Humalog, Eli Lilly).

## Immunofluorescence

Following euthanasia, collected pancreata were processed as described (*Desgraz and Herrera, 2009*). Paraffin and cryostat sections were 5 and 10 µm-thick, respectively. Primary antibodies: guinea pig anti-insulin (1:400, Dako, Denmark), mouse anti-glucagon (1:250 to 1:1000, Sigma), and rabbit anti-GFP (1:200, Molecular Probes Inc., Eugene, OR). Secondary antibodies were coupled to Alexa Fluor dyes 488, 568, or 647 (1:500, Molecular Probes Inc.); or to FITC, Cy3, or Cy5 (1:500, Jackson ImmunoResearch, West Grove, PA). Images were acquired on a confocal microscope (TCS SPE, Leica Microsystems, Germany). For cell mass measurement, 8 to 12 equally spaced sections per pancreas were imaged on a Leica M205 FA stereo microscope. Islets were manually selected using ImageJ (NIH) and thresholding was applied to measure the insulin- and glucagon-positive areas.

## RNA extraction and RT-qPCR

After dissection, liver and skeletal muscle (gastrocnemius) were immediately stored in RNAlater (Sigma). Tissues were homogenized with a Polytron and total RNA was extracted with the Qiagen (Germany) RNeasy mini kit (standard kit for liver, fibrous tissue kit for muscle). Reverse transcription was performed using the Qiagen QuantiTect RT kit. qPCR reactions and analyses were performed as described (*Thorel et al., 2010*); each sample was run in triplicate. For normalization, eight housekeeping genes were tested and the three more stable across our experimental conditions were

defined using geNorm (*Vandesompele et al., 2002*): *β-Glucuronidase (Gusb)*, *Glyceraldehyde-3-phosphate dehydrogenase (Gapdh)*, and *Non-POU-domain-containing, octamer binding protein (Nono)* for liver; *β-actin (Actb)*, *Gapdh*, and *Gusb* for skeletal muscle. Primer sequences are indicated in *Supplementary file 1*.

## Hormone and glycogen measurements

Protein extracts from total pancreas were prepared as described (*Strom et al., 2007*). Blood samples were collected in EDTA-coated tubes and plasma was separated by centrifugation. Insulin and glucagon concentrations were measured using Ultrasensitive Mouse Insulin and Glucagon ELISA kits (Mercodia, Sweden), respectively. Glycogen concentration was measured from the supernatatant of homogenized liver tissue using a glycogen asssay kit (Sigma).

## Immunoblotting

Liver samples were lyzed in radioimmumoprecipitation (RIPA) buffer with protease inhibitors (Thermo Fisher Scientific, Waltham, MA). Protein concentration was measured using a BCA assay (Thermo Fisher Scientific). Proteins were resolved on a TruPAGE gel (Sigma) and transferred to a PVDF membrane. The membrane was blocked in Tris-buffered saline with 0.1% Tween containing 5% bovine serum albumin. Primary antibodies were rabbit anti-PEPCK (1:1500, Abcam, UK) and mouse anti-tubulin (1:2500), both incubated overnight at 4°C; secondary antibodies were horseradish peroxidase-conjugated anti-rabbit (1:5000) and anti-mouse (1:5000). Proteins were detected using ECL plus substrate (Thermo Fisher Scientific) and images were acquired on a LAS-4000 imager (Fujifilm, Japan).

## Statistical analyses

Data are presented as mean ± SEM. $P$ values were calculated with GraphPad Prism 6 (GraphPad Software, La Jolla, CA). The following statistical tests were applied: unpaired, two-tailed, Mann-Whitney $U$ test for two sample comparisons; one- or two-way ANOVA with post hoc Bonferroni correction for multiple comparisons; Log-rank (Mantel-Cox) test for survival analyses.

# Acknowledgements

We thank Gissela Cabrera Gallardo, Carine Gysler and Muriel Urwyler for excellent technical help. We thank Rohn Millican and Paul Cain for generating Gcgr Ab reagent, and Lauge Schäffer (Novo Nordisk) for kindly providing S961. Work was funded by grants from the Department of Veterans Affairs, the NIH (DK66636, DK72473, DK89572, DK89538), the Vanderbilt Diabetes Research and Training Center (DK20593), and the Juvenile Diabetes Research Foundation (JDRF) (to ACP), as well as the Institute of Genomics and Genetics of Geneva (iGE3), the Swiss National Science Foundation (National Research Programme NRP63), the NIH (Beta Cell Biology Consortium), the JDRF, and the European Union (to PLH).

# Additional information

## Competing interests

JSM: Employee and shareholder of Eli Lilly and Company. The other authors declare that no competing interests exist.

## Funding

| Funder | Grant reference number | Author |
| --- | --- | --- |
| Institute of Genomics and Genetics of Geneva | | Nicolas Damond Pedro L Herrera |
| Juvenile Diabetes Research Foundation | | Alvin C Powers Pedro L Herrera |
| U.S. Department of Veterans Affairs | | Alvin C Powers |

| National Institutes of Health | DK66636 | Alvin C Powers |
|---|---|---|
| Vanderbilt Diabetes Research and Training Center | DK20593 | Alvin C Powers |
| National Institutes of Health | DK72473 | Alvin C Powers |
| National Institutes of Health | DK89572 | Alvin C Powers |
| National Institutes of Health | DK89538 | Alvin C Powers |
| Schweizerischer Nationalfonds zur Förderung der Wissenschaftlichen Forschung | NRP63 | Pedro L Herrera |
| National Institutes of Health | BCBC & HIRN | Pedro L Herrera |
| European Union | Imidia | Pedro L Herrera |

The funders had no role in study design, data collection and interpretation, or the decision to submit the work for publication.

### Author contributions
ND, Conceived and performed the experiments and analyses, Wrote the manuscript; FT, PLH, Conceived the experiments, Analysis and interpretation of data, Wrote the manuscript; JSM, Shared the anti-Gcgr mAb and contributed to discussion, Contributed unpublished essential data or reagents; MJC, Generated and shared the Gcgr-/- mice, Contributed to the planning of experiments, Edited the manuscript, Contributed unpublished essential data or reagents; PMV, Contributed unpublished essential data or reagents, Generated and shared the Gcgr-/- mice, contributed to the planning of experiments and edited the manuscript; ACP, Generated and shared the Gcgr-/- mice, Contributed to the planning of experiments, Edited the manuscript, Conception and design, Contributed unpublished essential data or reagents

### Author ORCIDs
Nicolas Damond, http://orcid.org/0000-0003-3027-8989
Pedro L Herrera, http://orcid.org/0000-0003-0771-9504

### Ethics
Animal experimentation: All mice were housed and treated in accordance with the guidelines and regulations of the Direction Générale de la Santé, state of Geneva (license number GE/103/14).

## Additional files

### Supplementary files
• Supplementary file 1. Primer sequences used for RT-qPCR.

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
