## [Decision Letter]

Thank you for submitting your work entitled "Blockade of glucagon signaling prevents or reverses diabetes onset only if residual β-cells persist" for consideration by *eLife*. Your article has been reviewed by three peer reviewers, and the evaluation has been overseen by Guy Rutter as the Reviewing Editor and Fiona Watt as the Senior Editor.

The reviewers have discussed the reviews with one another and the Reviewing Editor has drafted this decision to help you prepare a revised submission.

The following individuals involved in review of your submission have agreed to reveal their identity: Gordon Weir (peer reviewer).

Summary:

The manuscript is timely since the paper from Unger et al. (2011), which the present manuscript challenges, suggested that suppression of glucagon action alone (in the absence of insulin signalling) is sufficient to prevent diabetes, giving strong support to the "bihormonal" model of diabetes induction. This work from Herrera and colleagues demonstrates (though doesn't quite prove) that the model used by Unger and colleagues was likely to involve persistent insulin production and signalling, rather weakening this assertion. The other observation here – that α cells can still convert to β cells in the absence of glucagon signalling – is also interesting, but is arguably tangential to the main thrust of the paper.

The paper is well written and of sound scientific quality. The subject is interesting and valuable for a broad range of researchers. Importantly, it clarifies an existing puzzle in the literature and has therapeutic implications.

In summary, this is an important paper for the field. However, there are important suggestions for revision, including discussion of recent relevant literature (notably the recent report by JJ Holst and colleagues in Diabetologia), reanalysis of β and α cell mass data and/or possible further experimentation to provide adequate quantification of these parameters.

Essential revisions:

1) The ablation of β cells by STZ was certainly incomplete as is often the case. It is not easy to determine the perfect dose, but more complete ablation might have been achieved with a higher dose and a longer fast, which would have lowered the glucose level and enhanced the STZ uptake by the β cells. In any case, it would be helpful to explain in the Introduction that STZ treatment does not always destroy all the β-cells in the pancreas. It depends on the dose, mouse strain, etc.

In particular, the authors need to explain how they chose their STZ concentrations. Even the lowest dosage they used (175mg/kg) is almost twice as high that as in the cited papers (100mg/kg). This lower dose likely explains why ablation of glucagon signalling was beneficial in previous studies, and this should be discussed. In any case, the STZ concentration used should be stated in the legend of each figure, given that very small changes can affect β-cell loss and thus the effect of glucagon signalling ablation. (e.g. in Figure 3,Figure 4)

2) Related to this, the pancreatic insulin content of the STZ group was 21% of normal. If the glucose was high there might have been degranulation, so is it possible that the actual β cell mass was even higher, that is over 21% of normal.

3) Regarding the pancreatic insulin content. If a normal mouse pancreas weighs 100 mg, the insulin content would be 100 ng x 100 mg = 10,000 ng or 10 μg per mouse panaceas, which is about what others get. In Figure 6, after the DT, the insulin content of the pancreases is 10 ng/pancreas, which show the remarkable efficiency of DT ablation. In this figure please indicate the significance level for the asterisks shown.

4) There is great interest in the mass of β cells required to prevent diabetes in both mice and humans. The current paper does not have true mass measurements. It is difficult to make sense of pancreatic insulin content and number of β cells per islet section. The paper would be stronger if mass were measured in some groups of the mice used in this study. In particular, in would be of great interest to know the β cell mass required to prevent diabetes with and without glucagon receptors either knocked out or blocked. Related to this, it would also be of value to determine the mass of α cells and the mass of insulin/glucagon double positive cells. This gets at the important question of understanding of how many β cells can be generated from α cells and whether they make a meaningful contribution to insulin secretion.

5) A recent paper in Diabetologia (vol 59, p363) from Holst and colleagues comes to a similar conclusion although with a somewhat different approach. Glucagon immunoneutralisation was tested, with the same results as reported here. However, glucagon signalling was also disrupted by ablation of glucagon-secreting α cells or using a glucagon receptor antibody. This paper should be discussed. Even older work from the same laboratory (PMID: 7851693) is also relevant here and should be cited.

6) Although the authors show measures of blood-glucose levels which is the final output of interest, the mechanistic approach in the attempt to explain the reasons for why insulin is needed is lacking. The measurements on the important liver enzymes is limited and only mRNA-levels are evaluated. At least the key glucogenic protein PEPCK should also be measured on protein level.

7) In addition, gene expression levels of other key lever genes could be measured (e.g. G6Pc, Fbp1 and Pcx) to get a more detailed picture on the liver function.

---

## [Author Response]

Essential revisions:

1) The ablation of β cells by STZ was certainly incomplete as is often the case. It is not easy to determine the perfect dose, but more complete ablation might have been achieved with a higher dose and a longer fast, which would have lowered the glucose level and enhanced the STZ uptake by the β cells. In any case, it would be helpful to explain in the Introduction that STZ treatment does not always destroy all the β-cells in the pancreas. It depends on the dose, mouse strain, etc.

STZ-mediated β-cell ablation was indeed incomplete, as clearly shown by our data in Figure 3. Although altering the STZ injection protocol may have slightly increased the ablation efficiency, it is very unlikely that we could have achieved a near-total β-cell destruction in glucagon receptor knockout (*Gcgr-/-*) animals, which are less sensitive to this chemical than their *Gcgr+/+* or *Gcgr+/-* counterparts. This is precisely why we opted for alternative methods: i) the more efficient DT-mediated ablation in *RIP-DTR;Gcgr-/-* mice; ii) additional insulin signaling inhibition with S961 after STZ treatment; and iii) glucagon signaling blockade in STZ-treated C57BL/6 mice.

We have added a sentence in the Introduction to indicate that the efficiency of STZ-mediated β-cell ablation depends on the administration protocol and mouse strain.

In particular, the authors need to explain how they chose their STZ concentrations. Even the lowest dosage they used (175mg/kg) is almost twice as high that as in the cited papers (100mg/kg). This lower dose likely explains why ablation of glucagon signalling was beneficial in previous studies, and this should be discussed. In any case, the STZ concentration used should be stated in the legend of each figure, given that very small changes can affect β-cell loss and thus the effect of glucagon signalling ablation. (e.g. in Figure 3,Figure 4)

We performed pilot experiments to determine optimal STZ dosages (maximization of β-cell ablation with low mortality). Please note that in the cited papers, 100 mg/kg STZ was administered i.v., whereas we used a higher dose injected i.p.

We used two different protocols of STZ administration, depending on the genetic background:

a) *Gcgr+/-* and *Gcgr-/-* mice (Figure 3 and Figure 4): first injection of 200 mg/kg followed one week later by a second injection of 150 mg/kg.

b) C57BL/6 mice (Figure 5): single injection of either 175 mg/kg or 225 mg/kg.

We added the STZ administration protocol to the legend of each figure and clarified the STZ paragraph in the Materials and methods section.

Regarding the beneficial effect of glucagon signaling, we reach the same outcome as the mentioned papers, that is: *Gcgr-/-* mice remain normoglycemic after STZ treatment. Here, we report that if a more efficient method of β-cell ablation (DT-mediated) or additional insuling signaling blockade (STZ+S961) are applied, *Gcgr-/-* animals become severely diabetic despite the complete absence of glucagon signaling.

2) Related to this, the pancreatic insulin content of the STZ group was 21% of normal. If the glucose was high there might have been degranulation, so is it possible that the actual β cell mass was even higher, that is over 21% of normal.

As reported in previous papers and confirmed by us in this manuscript (see Figure 4, purple inverted triangles), *Gcgr-/-* mice remain normoglycemic after treatment with STZ.

To assess degranulation, we stained islets from untreated or STZ-treated *Gcgr-/-* mice for Nkx6.1 and insulin. As shown in the pictures below, over 90% of Nkx6.1-positive cells still co-expressed insulin after STZ, arguing against a significant β-cell degranulation in these animals.

3) Regarding the pancreatic insulin content. If a normal mouse pancreas weighs 100 mg, the insulin content would be 100 ng x 100 mg = 10,000 ng or 10 μg per mouse panaceas, which is about what others get. In Figure 6, after the DT, the insulin content of the pancreases is 10 ng/pancreas, which show the remarkable efficiency of DT ablation. In this figure please indicate the significance level for the asterisks shown.

DT-mediated β-cell ablation is very efficient indeed. Our values for pancreatic insulin content are consistent with the estimates of this reviewer: 150 ng/mg in untreated and 0.1 ng/mg in DT-treated *RIP-DTR;Gcgr+/-* mice (see Figure 3—figure supplement 2, new data provided in the revised manuscript).

We have added the requested P values directly on the graphs (Figure 6=0.022 and Figure 6=0.015).

4) There is great interest in the mass of β cells required to prevent diabetes in both mice and humans. The current paper does not have true mass measurements. It is difficult to make sense of pancreatic insulin content and number of β cells per islet section. The paper would be stronger if mass were measured in some groups of the mice used in this study. In particular, in would be of great interest to know the β cell mass required to prevent diabetes with and without glucagon receptors either knocked out or blocked. Related to this, it would also be of value to determine the mass of α cells and the mass of insulin/glucagon double positive cells. This gets at the important question of understanding of how many β cells can be generated from α cells and whether they make a meaningful contribution to insulin secretion.

We measured the β- and α-cell masses in untreated, STZ- and DT-treated RIP-*DTR;Gcgr+/+* and *RIP-DTR;Gcgr-/-* mice. These new data were added to the Figure 3 in the revised manuscript (Figure 3, in replacement of previous panel 3B, and Figure 3—figure supplement 1 and Figure 3—figure supplement 2). Of note, we find a good correlation between the β-cell mass and the pancreatic insulin content.

The question of the β-cell mass required to prevent diabetes in the presence or absence of glucagon signaling is very interesting. However, differences in sensitivity to STZ between *Gcgr+/-* and *Gcgr-/-* mice makes it difficult to obtain a similar ablation efficiency in these two genotypes. A large cohort of mice treated with different doses of STZ combined or not with anti-GCGR mAb administration would be required to accurately evaluate the minimal β-cell mass needed to prevent diabetes in situations of normal or blocked glucagon signaling. In this study, we already observe a beneficial effect of glucagon receptor blockade therapy on the glycemia of STZ-treated C57BL/6 mice in presence of a tiny amount of residual insulin (1.79% of pancreatic insulin content, Figure 5).

The bihormonal cell mass is difficult to measure, but we can calculate it indirectly. After DT in *RIP-DTR;Gcgr-/-* animals, we find that the average α-cell mass is 4.92 mg (Figure 3—figure supplement 1) and that 0.73% of glucagon-positive cells make insulin (Figure 6). Thus, we can estimate the bihormonal cell mass to be around 36 μg, that is less than 3% of the β-cell mass in untreated mice. Even if these cells were secreting insulin like β-cells, this amount would be insufficient to prevent hyperglycemia. Rescuing insulin deficiency by conversion of α-cells will therefore require additional interventions to promote α-cell reprogramming. This however falls outside of the scope of this manuscript.

5) A recent paper in Diabetologia (vol 59, p363) from Holst and colleagues comes to a similar conclusion although with a somewhat different approach. Glucagon immunoneutralisation was tested, with the same results as reported here. However, glucagon signalling was also disrupted by ablation of glucagon-secreting α cells or using a glucagon receptor antibody. This paper should be discussed. Even older work from the same laboratory (PMID: 7851693) is also relevant here and should be cited.

We have modified the Discussion to include this very recent publication.

In agreement with this study, we also reported that α-cell ablation does not improve glycemic control after massive DT-mediated α- and β-cell co-ablation in *RIP-DTR;Glucagon-DTR* double transgenic mice (Thorel et al., Diabetes, 2011).

6) Although the authors show measures of blood-glucose levels which is the final output of interest, the mechanistic approach in the attempt to explain the reasons for why insulin is needed is lacking. The measurements on the important liver enzymes is limited and only mRNA-levels are evaluated. At least the key glucogenic protein PEPCK should also be measured on protein level.

We performed a western blot analysis to measure Pepck expression at the protein level in untreated and DT-treated *RIP-DTR;Gcgr+/-* and *RIP-DTR;Gcgr-/-* mice. This data was added to the revised manuscript (Figure 4—figure supplement 1). Consistent with mRNA data, PEPCK protein expression is higher in DT-treated animals.

7) In addition, gene expression levels of other key lever genes could be measured (e.g. G6Pc, Fbp1 and Pcx) to get a more detailed picture on the liver function.

We have chosen to show the expression levels of Pepck and Gck because the expression of these genes is known to be regulated by both glucagon and insulin at the transcriptional level.

As requested, we measured the liver expression of G6pc, Fbp1, Pcx, and Pklr in our different experimental conditions. G6pc, a key enzyme in gluconeogenesis and glycogenolysis, displayed a similar regulation as that of Pepck consistent with increased glucose mobilization after DT.

The other genes displayed a subtler regulation in response to alterations of insulin (Pcx) or glucagon (Pklr, Fbp1) action. As it is more difficult to draw conclusions from these small variations, we have not included these data in the revised manuscript.

Altogether, the results provided in Figure 4 show that after DT in *Gcgr-/-* animals: i) the expression of key gluconeogenic genes in the liver is increased (Pepck and G6pc at the transcript level, PEPCK at the protein level), ii) the liver glycogen content is reduced, and iii) the expression of insulin-sensitive genes, such as Irs2, is increased in skeletal muscle. Combined, these data support the hypothesis that DT treatment causes a severe insulin deficiency that cannot be compensated by the absence of glucagon signaling in these animals. This likely leads to a higher glucose mobilization and/or a lower glucose uptake resulting in hyperglycemia and diabetes.